# The Effects of *DDI1* on Inducing Differentiation in Ovine Preadipocytes via Oar-miR-432

**DOI:** 10.3390/ijms241411567

**Published:** 2023-07-17

**Authors:** Meilin Jin, Zehu Yuan, Taotao Li, Huihua Wang, Caihong Wei

**Affiliations:** 1Institute of Animal Sciences, Chinese Academy of Agricultural Sciences, Beijing 100193, China; jmlingg@163.com (M.J.); ltt_ltt2020@163.com (T.L.); 2College of Animal Science and Technology, China Agricultural University, Beijing 100193, China; 3College of Animal Science and Technology, Yangzhou University, Yangzhou 225009, China; yuanzehu@yzu.edu.cn

**Keywords:** oar-miR-432, *DDI1*, ovine preadipocyte, fat deposition

## Abstract

Reducing fat deposition in sheep (*Ovis aries*) tails is one of the most important ways to combat rising costs and control consumer preference. Our previous studies have shown that oar-miR-432 is differentially expressed in the tail adipose tissue of Hu (a fat-tailed sheep breed) and Tibetan (a thin-tailed sheep breed) sheep and is a key factor in the negative regulation of fat deposition through *BMP2* in ovine preadipocytes. This study investigated the effect of oar-miR-432 and its target genes in ovine preadipocytes. A dual luciferase assay revealed that *DDI1* is a direct target gene of oar-miR-432. We transfected an oar-miR-432 mimic and inhibitor into preadipocytes to analyze the expression of target genes. Overexpression of oar-miR-432 inhibits *DDI1* expression, whereas inhibition showed the opposite results. Compared with thin-tailed sheep, *DDI1* was highly expressed in the fat-tailed sheep at the mRNA and protein levels. Furthermore, we transfected the overexpression and knockdown target genes into preadipocytes to analyze their influence after inducing differentiation. Knockdown of *DDI1* induced ovine preadipocyte differentiation into adipocytes but suppressed oar-miR-432 expression. Conversely, the overexpression of *DDI1* significantly inhibited differentiation but promoted oar-miR-432 expression. *DDI1* overexpression also decreased the content of triglycerides. Additionally, *DDI1* is a nested gene in intron 1 of *PDGFD*. When *DDI1* was overexpressed, the *PDGFD* expression also increased, whereas *DDI1* knockdown showed the opposite results. This is the first study to reveal the biological mechanisms by which oar-miR-432 inhibits preadipocytes through *DDI1* and provides insight into the molecular regulatory mechanisms of *DDI1* in ovine preadipocytes. These results have important applications in animal breeding and obesity-related human diseases.

## 1. Introduction

The earliest records of fat-tailed sheep date back to about 5000 years ago, and it is believed that fat-tailed sheep evolved from the domestication of thin-tailed sheep, with them representing approximately 25% of the world’s sheep population [1]. The formation of fat tails was an adaptive response mechanism to the rapid decline in the Earth’s climate [1,2]. Fat-tailed sheep store fat in their tail and buttocks, which accounts for up to 20% of their carcass weight [3]. Excessive deposition of fat in the tail affects reproduction and increases feeding costs. Therefore, farmers prefer to breed thin-tailed sheep. Meanwhile, the phenotypic diversity of sheep tail types provides ideal material for comparative analysis. Animal geneticists and breeders have always wanted to clearly understand the underlying genetic mechanisms behind the differences in the phenotype of sheep tails [4]. Understanding the causal genes and mutations behind these differences will help explore evolutionary mysteries, improve animal performance, and provide insights into human diseases associated with fat deposition (i.e., obesity).

Genomics and transcriptomics provide an unprecedented opportunity to understand the mechanisms of tail fat formation in sheep and identify potential causal variations between phenotypic differences. Up until now, researchers have conducted genomic and transcriptomic studies into the mechanisms of fat deposition in sheep tails. Several genes have been identified as potentially important candidates, including bone morphogenetic protein 2 (*BMP2*) [5], DNA damage inducible 1 homolog 1 (*DDI1*) [6] and platelet-derived growth factor D (*PDGFD*) [4,6], which are associated with fat-tail phenotypes. Among these genes, a previous study showed that *BMP2* promotes the expression of *PPAR-γ* in preadipocytes and promotes the differentiation of preadipocytes from sheep [5,7]. *DDI1* is a eukaryotic protein with a retroviral protease fold [8,9], which has been previously identified both in Drosophila photoreceptor neurons and in human neuroblastoma cells in culture as a direct substrate of the ubiquitin E3 ligase [10,11]. However, no studies have explained the molecular mechanisms of *DDI1* and how they regulate fat deposition in sheep tails. *PDGFD* encodes a member of the platelet-derived growth factor family. Studies have shown that *PDGFD* plays a critical role in inhibiting adipogenesis by suppressing preadipocyte differentiation and inhibiting fat development [12]. A series of research works found *PDGFD* to be a high-ranking candidate gene for the sheep fat-tail phenotype [4]. A single-nucleotide polymorphism (SNP) mutation site and allele-specific expression (ASE) were found in the first exon [13]. A 6.8kb region containing the most differentiated SNPs in intron 1 was also positively selected [14]. Transcript I of *PDGFD* was found to be the most differentially expressed isoform between fat-tailed and thin-tailed sheep [15]. Although potential genes associated with sheep tail characteristics have been revealed, causal variations and regulatory mechanisms in these potential candidate genes remain elusive and require further research. 

Furthermore, microRNA (miRNA) is a kind of non-coding RNA with a length of about 22nt. MiRNA is a major class of gene regulators involved in many biological processes via negatively regulating mRNAs [16,17]. MiRNAs direct silencing complexes to either degrade RNA or inhibit mRNA translation by binding to complementary target gene RNA sequences. The seed region of miRNA combines with the 3′UTR of genes to induce degradation or inhibit target gene translation [18]. In fat development, various miRNAs have been discovered to have crucial roles in adipogenesis, such as miR-133 [19,20], miR-29a [21], miR-122 [22,23,24] and miR-34 [25,26]; miR-379-5p has been associated with tail fat deposition in sheep [27]. In these studies, miR-432 inhibited milk fat synthesis in sheep mammary epithelial cells [28], and it was significantly down-regulated in the fat-tailed sheep breed. Oar-miR-432 regulates fat differentiation and promotes the expression of *BMP2* in ovine preadipocytes [5]. The expression pattern of oar-miR-432 in fat-tailed sheep has been studied, but a complete model of its downstream regulation functions has not been established. Accordingly, in this study, we investigated the effects of oar-miR-432 on the expression of the target gene. We also analyzed the effect of target genes on the expression of oar-miR-432 and the content of triglycerides in ovine preadipocytes.

## 2. Results

### 2.1. Oar-miR-432 Is Down-Regulated and Differentially Expressed in Sheep Tail Fat 

To identify important miRNAs for tail fat differentiation in fat-tailed sheep, we previously analyzed miRNA-seq in the tail adipose tissue of fat-tailed sheep and thin-tailed sheep [27]. Among these DE miRNAs, oar-miR-432 (FPKM = −2.66, Q-value = 1.86 × 10^−5^) was the most down-regulated in fat-tailed sheep (Appendix A). Potential target genes were predicted by RNAhybrid, miRanda, and TargetScan software. 

There are three potential target genes (*BMP2*, homeobox A3 (*HOXA3*), and *DDI1*) as determined by a previous selection sweep and miRNA-seq [5,6]. Among these genes, it has been verified that *BMP2* and *HOXA3* play significant roles in fat tail formation. Based on our previous studies, *BMP2* is the peak gene harbored in the largest region identified by hapFLK [6]. Additionally, oar-miR-432 inhibited fat deposition and promoted the expression of *BMP2* in ovine preadipocytes [5]. *HOXA3* also has potential roles in body formation [6]. *DDI1* was down-regulated in fat-tailed sheep at the mRNA and protein level (Figure 1A–C) and has a strong selection signature in the fat-tailed sheep genome [6], which led us to hypothesize that natural selection may have had a role in aspects of adipogenesis stimulation. Based on these results, *DDI1* was selected for verification using the dual luciferase reporter assay.

### 2.2. Oar-miR-432 Inhibits Fat Deposition and Directly Targets DDI1

To test this hypothesis, *DDI1* was selected for verification using a dual luciferase reporter assay. The 3′ UTR segments of *DDI1* were cloned into the psiCHECK2 luciferase reporter construct, which also had the predicted oar-miR-432 target site or mutated seed sites (Figure 2A). The binding of oar-miR-432 to the 3′UTR of wild-type *DDI1* was verified by a luciferase assay in HEK293T cells (Figure 2B). This result demonstrates that oar-miR-432 directly interacts with the predicted target sites in *DDI1*. The over-expression of oar-miR-432 led to a decrease in the expression levels of the fat synthesis marker genes, *FABP4* and *ACACA*, after four days (*p* < 0.01, Figure 2C,D). These results showed that oar-miR-432 negatively regulates fat deposition in preadipocytes.

### 2.3. Oar-miR-432 Negatively Regulates the Expression of DDI1 in Ovine Preadipocytes

Subsequently, we investigated the involvement of *DDI1* in the regulation of oar-miR-432 during ovine preadipocyte differentiation. We transfected the oar-miR-432 mimic into ovine preadipocytes. The expression level of oar-miR-432 was increased more than five-fold in preadipocytes after two and four days [5]. Strikingly, after two days, oar-miR-432 overexpression decreased the expression level of *DDI1* when compared with the negative control (NC) (Figure 3A). There was no significant difference at day four, but the trend was the same as at day two. We also verified the DDI1 protein expression at two and four days, and the results were consistent with the mRNA expression level (Figure 3C,D). However, there was no significant difference after two days. In contrast, the inhibition of oar-miR-432 markedly increased *DDI1* mRNA expression levels after two and four days (Figure 3B). These results demonstrated that oar-miR-432 regulates fat differentiation by inhibiting *DDI1* expression in ovine preadipocytes.

### 2.4. DDI1 Influences the Expression of Oar-miR-432 in Ovine Preadipocytes

Oar-miR-432 was found to negatively regulate the expression of *DDI1* in ovine preadipocytes; thus, we next explored the role of *DDI1* in the regulation of oar-miR-432. Lentivirus-induced *DDI1* overexpression or *DDI1* knockdown was transferred into preadipocytes, with the success rate of lentivirus transfection reaching 90% (Figure 4A,D). *DDI1* also enhanced the expression after transfection of the lentivirus-induced overexpression of *DDI1* (Figure 4B) and suppressed the expression after transfer of the lentivirus-induced *DDI1* knockdown (Figure 4E). These results showed that lentiviral *DDI1* overexpression and *DDI1* knockdown were successfully transfected into the preadipocytes. Furthermore, treatment of preadipocytes with induction differentiation markedly decreased the expression of oar-miR-432 (Figure 4C). This inhibition increased the expression of oar-miR-432 (Figure 4F). Together, these findings suggest that *DDI1* may influence the expression of oar-miR-432 after inducing differentiation in ovine preadipocytes. 

### 2.5. DDI1 Inhibits Ovine Preadipocytes Inducing Differentiation

Furthermore, to further verify *DDI1*’s regulation of fat deposition in ovine preadipocytes, we used a lentivirus to over-express or knockdown *DDI1* (Figure 4C,F). We further used the fat deposition marker gene peroxisome proliferator-activated receptor gamma (PPAR-γ) to verify *DDI1*’s regulation of fat deposition in preadipocytes. As expected, compared with the green fluorescent protein control, *DDI1* overexpression significantly suppressed the *PPAR-γ* expression (Figure 5A). A Western blot analysis showed that the protein expression of *PPAR-γ* was also higher than NC, which was consistent with the mRNA expression level (Figure 5B,C). Conversely, *DDI1* inhibition had the opposite effect on the mRNA level (Figure 5D) and protein level (Figure 5E,F). We also used the expression of the fat synthesis marker genes *FABP4* and *ACACA* to verify the function of *DDI1* in preadipocytes. The results are consistent with *PPAR-γ*. The overexpression of *DDI1* decreased the expression levels of *FABP4* and *ACACA* (*p* < 0.01, Appendix A). Furthermore, ovine preadipocytes were stained with oil red O. Many small lipid droplets were stained red, and lipid rings were visible. The number of lipid drops in the *DDI1* overexpressing group was lower than that of the NC group (Figure 5G). Compared with NC, *DDI1* overexpression increased the 2.17-fold-change differentiation rate in preadipocytes (Figure 5G), which showed that fat deposition in sheep tail is reduced by *DDI1* overexpression. These results indicate that *DDI1* negatively regulates the fat deposition in ovine preadipocytes.

### 2.6. DDI1 Co-Expression Patterns with PDGFD

In addition, *DDI1* is a nested gene and is located in intron 1 of *PDGFD* (Appendix A). *PDGFD* plays a crucial role in the fat deposition of fat-tailed sheep [6]; thus, we next investigated the influence of *DDI1* overexpression or knockdown on the expression of *PDGFD* in ovine preadipocytes. *PDGFD* levels increased with *DDI1* overexpression (Figure 6A), while *DDI1* knockdown had the opposite effect (Figure 6D) at the mRNA level. At the protein level, PDGFD showed the same expression trend as DDI1, but the difference was not significant (Figure 6B,C,E,F). Together, these data suggest that *DDI1* has co-expression patterns with *PDGFD* at the mRNA level in ovine preadipocytes. 

## 3. Discussion

At the post-transcriptional level, miRNAs are a significant class of gene regulators and are essential for preadipocyte proliferation and differentiation in many organisms [29]. MiRNA-seq was used to identify potential miRNAs related to sheep fat deposition in fat-tailed and thin-tailed sheep breeds [27]. Oar-miR-432 was down-regulated in fat-tailed sheep and promoted the expression of *BMP2* in ovine preadipocytes [5]. Additionally, oar-miR-432 may target a series of genes to regulate adipogenesis [28]. In this study, we have verified that oar-miR-432 directly targets and negatively regulates *DDI1* expression in ovine preadipocytes. *DDI1* also down-regulates oar-miR-432 in ovine preadipocytes. We found in our previous study that *DDI1* has a strong selection signature in fat-tailed sheep breeds [6]. Variation in the *PPAR-γ* expression has been reported to be positively correlated with fat deposition [30]. The expression of the fat deposition marker gene *PPAR-γ* was significantly enhanced when *DDI1* was knocked-down in preadipocytes. *DDI1* overexpression had the opposite effect. Significantly, we demonstrated that *DDI1* inhibits fat differentiation in preadipocytes. This study provides insights into the fat regulation mechanism in fat-tailed sheep and has important applications in animal breeding as well as obesity-related human diseases. In humans, when fat accumulates disproportionately in the viscera rather than in the buttocks and hips, the risks of cardiovascular metabolic diseases such as coronary heart disease are increased [31].

Meanwhile, research on *DDI1* has concentrated on proteolytic functions during regulated protein turnover in cells, but no specific mechanism of fat deposition has been elucidated [8]. *DDI1* functions as a substrate receptor of the CUL4-DDB1 ubiquitin ligase [32]. *DDB1* is widely recognized as an important protein in the Cullin 4 (CUL4) ubiquitin ligase complex, which regulates a range of physiological processes by recognizing and ubiquitinating substrate proteins containing the WD40 domain [33]. Previous studies have shown that *DDB1* as a transcription factor directly binds *UCP1* and *PPARGC1A* promoters, thereby enabling rapid and synchronized transcription of thermogenic genes upon acute cold exposure [34]. WD and tetratricopeptide repeat 1 (WDTC1) serve as substrate receptors for DDB1-CUL4-ROC1 (CRL4) E3 complexes, which are involved in transcriptional repression during adipogenesis [35]. While reduced *DDI1* expression is associated with fat deposition in preadipocytes. Whether *DDI1* affects fat deposition by forming complexes with *DDB1* such as *WDTC1* needs to be verified by subsequent experiments. 

Meanwhile, *DDI1* is a nested gene and is located in intron 1 of *PDGFD*, which has a strong selection signature in Chinese and Mediterranean fat-tailed sheep [14,15,36]. A previous study has shown that several genes are complex and overlap, and these arrangements potentially contribute to the regulation of gene expression. There are co-expression patterns among these overlapping genes [37]. However, there are no studies on how the molecular mechanisms of *DDI1* and *PDGFD* regulate adipogenesis in sheep tails. Herein, we also verified whether *PDGFD* has co-expression patterns with *DDI1*. As anticipated, the expression of *PDGFD* increases when *DDI1* is overexpressed and decreases when *DDI1* is knocked down in preadipocytes. We speculate that the overlapping genes *PDGFD* and *DDI1* may play a key role in the occurrence and development of adipogenesis. Further research is essential to accurately elucidate the mechanical architecture of this gene in adipogenesis.

Moreover, there have been reports that miR-432 inhibits milk fat synthesis and the proliferation and differentiation of bovine primary myoblasts by inhibiting the IGF2/PI3K pathway [38]. A previous study has also shown that miR-432 can target and negatively regulate stearoyl-CoA desaturase (*SCD*) and lipoprotein lipase (*LPL*) [28]. Additionally, miR-432 overexpression inhibits *FABP4* (fatty acid binding protein 4, adipocyte), lipin 1 (*LPIN1*) and acetyl-CoA carboxylase alpha (*ACACA*) expression [28]. The genes *FABP4* and *ACACA* are crucial in the synthesis of triglycerides. *FABP4* can bind to long-chain fatty acids and deliver them to the receptor [39]. *ACACA* is a key rate-limiting enzyme, and catalyzes acetyl-CoA to produce malonyl-CoA [40]. Acyl-CoA is then produced using long-chain fatty acids and de novo synthesized fatty acids in preadipocytes [41]. In our study, the oar-miR-432 mimic decreased the expression of *FABP4* and *ACACA*. These results demonstrate the negative regulatory role of oar-miR-432in fat deposition. Meanwhile, the overexpression of *DDI1* also decreased the expression of *PPAR-γ*, *FABP4,* and *ACACA*, and oar-miR-432 targets *DDI1*. In addition, Jiang et al. [42] found that miR-432 activated Wnt/β-catenin signaling when downregulated in human hepatocellular carcinoma cells, as well as via the maintenance of adult precursor cells in a pluripotent state [43]. It was therefore inferred that miR-432 may also inhibit fat deposition by regulating the Wnt/β-catenin signaling pathway. Meanwhile, previous studies have shown that the Wnt/β-catenin pathway plays a crucial role in adipocyte precursor cells. It may down-regulate *PPAR-γ* expression and inhibit adipocyte formation [44]. *BMP2* induces the directed differentiation of stem cells into white fat and influences fat deposition in sheep tail [5]. In humans, *BMP2* is a novel depot-specific regulator of adipogenesis in subcutaneous adipose tissue [45]. *BMP2* increases hyperplasia and hypertrophy of bovine subcutaneous preadipocytes via BMP/SMAD signaling [46]. These results suggest that *BMP2* may have similar regulatory mechanisms for fat deposition in different species. 

Accordingly, in conjunction with our own findings, we proposed a *DDI1* regulatory model of adipogenesis described in the following. Oar-miR-432 is regulated by *DDI1* and *BMP2* [5]. Oar-miR-432 may act as a decoy to relieve its stimulative effect on *BMP2*, which activates the Wnt/β-catenin pathway. We suspected that activation of the Wnt/β-catenin pathway influences *PPAR-γ* expression, eventually forming an auto-regulatory feedback loop during adipogenesis in sheep tails (Appendix A). We present a preliminary analysis of the mechanism of *DDI1* in tail fat deposition for the first time. 

## 4. Materials and Methods

### 4.1. Cell Isolation and Culture

Sheep primary preadipocytes and HEK293T (human embryonic kidney cells) were used in this study. Sheep primary preadipocytes were isolated from the tail adipose tissue of fat-tailed sheep (70-day-old fetus). Additionally, preadipocytes were cultured in DMEM medium supplemented with 10% fetal bovine serum (FBS, Gibco, Waltham, MA, USA) and 5% penicillin-streptomycin (PS, Gibco, Waltham, MA, USA). HEK293T cells were cultured in high glucose DMEM medium supplemented with 10% FBS and 5% PS. 

### 4.2. Prediction of Oar-miR-432 Target Genes and Verification by a Dual Luciferase Reporter Assay

MiRNA libraries of two sheep breeds (Hu and Tibetan sheep breed) with each of three biological replicates were sequenced using the BGISEQ-500 platform. The mapped clean tags aligned to the sheep reference genome (oar_v3.1). A total of 105 differentially expressed (DE) miRNAs (43 up-regulated and 62 down-regulated) were identified with FDR ≤ 0.01 and |FRKM| ≥ 1.5 by the DE-seq program [27]. The target genes of miRNAs were predicted by RNAhybrid [47], TargetScan [48] (http://www.targetscan.org) and miRanda (http://www.microrna.org/microrna/home.do) algorithms and verified by a dual luciferase reporter assay. The oar-miR-432 mimic and NC were designed and synthesized by GenePharam (Shanghai, China). The 3′UTR sequences for the target gene *DDI1* were PCR-amplified with specific primers (forward 5′-aattctaggcgatcgctcgagAACAGGCTCTATATATGCTG-3′, reverse 5′-attttattgcggccagcggccgcCATCCTGGGAGAAAAATTCA-3′) and cloned into the psiCHECK2 dual luciferase reporter vector as an XhoI/NotI insert. The oar-miR-432 seed motif is mutagenized at the 3′UTR of DDI1. The binding sites showed the following: 5′-tcctcttcggagatgctgATTGGAgatattataggatggcttgtccaa-3′ for mut1 and 5′-catcctataatatcTCCAATcagcatctccgaagaggatggttttc-3′ for mut2. An amount of 1 × 10^4^ HEK293T cells were cultured in 96-well plates in an incubator (5% CO_2_ and 37 °C). A total of 100 ng recombined vectors and 20 nM of either oar-miR-432 or NC were transfected into cells with opti-medium (Gibco, USA). The cell medium was then completely changed after 6–8 h [20]. After transfection for 48 h, HEK293T cells were collected for analysis with a Dual-Luciferase Reporter Assay System (Promega).

### 4.3. Cell Transfection

A total of 20 nM oar-miR-432 mimic, inhibitor, or NC at a density of 70% cells/well was inoculated into a 6-well plate for transfection in preadipocytes. When the density reached 100% cells/well, the cells were cultured in an induction differentiation medium for two days and changed to maintain induction differentiation medium for another two days. Lentivirus-induced *DDI1* overexpression or knockdown was transfected into preadipocytes in the same way as oar-miR-432. Lentivirus-induced *DDI1* overexpression, *DDI1* knockdown, and NC were constructed in GenePharam (Shanghai, China). Cells were cultured in 6-well plates in triplicate for 24 h and the titration of lentiviruses was multiplicity of infection (MOI) = 100 in the final construct, including *DDI1* overexpression or interference. When the cells showed contact inhibition, the induction differentiation medium was changed after two days. The induction differentiation medium contained 10% FBS, 5% PS, 1 µM dexamethasone (Macklin, Shanghai, China), and 0.5 mM isobutylmethylxanthine (Macklin, Shanghai, China) [49]. Finally, cells were cultured in maintenance differentiation medium for two days [49]. The maintained differentiation medium contained 10% FBS, 5% PS, and 10 mg/mL insulin (Macklin, Shanghai, China). Day one was set as the first day of culture after the introduction of the differentiation medium. Cells were collected from the oar-miR-432 mimic, inhibitor, overexpression of *DDI1,* and knockdown at various times (two, four days) to extract RNA and protein. 

### 4.4. Oil Red O Staining

Preadipocytes, which were maintained in induction differentiation medium, were removed from the 6-well cell culture plate, washed with PBS twice, and fixed with 4% paraformaldehyde for 30 min. Furthermore, the cells were stained with an oil red O solution (Solarbio, Beijing, China) for 30 min and then washed with 60% isopropanol. The cells were washed with PBS three times. Finally, the cells were visualized using a 40× microscope (Leica, Wetzlar, Germany) and the images were collected.

### 4.5. RT-qPCR

Cells were collected after transfection to extract RNA. Six fat tissue samples from sheep tails consisting of three Hu sheep (Qinghai, China) and three Tibetan sheep (Gansu, China) were used. All the samples were from adult ewes (six months old). Fat tissues were collected to extract RNA. The total RNA was isolated and purified using TRIzol (Invitrogen, Inc.) and an RNAprep Pure Micro Kit (TIANGEN, Beijing, China). A Nanodrop 2000 nucleic acid protein analyzer (Thermo, Waltham, MA, USA) was used to measure the purity and concentration of RNA. Reverse transcription was performed according to the instructions of the Prime Script II 1st strand kit (Takara, Beijing, China). The primer was designed by Primer 5.0 (Table 1). β-actin was the reference gene. Three biological replicates and triplicate technical replicates were obtained. 

Oar-miR-432 and the oar-5s reverse primer were designed in miRNA Design V1.01 software (Vazyme, Nanjing, China). MiRNA reverse transcription was conducted in accordance with the instructions of the miRNA 1st Strand cDNA Synthesis Kit (MR101-01/02, Vazyme) by stem-loop. A miRNA Universal SYBR qPCR Master Mix (MQ101-01/02, Vazyme) was used for miRNA qPCR. The qPCR reaction included one cycle at 95 °C for 5 min, followed by forty cycles at 95 °C for 10 s and 60 °C for 30 s, then one cycle at 95 °C for 15 s, 60 °C for 60 s, and 95 °C for 15 s. The accuracy of the data was analyzed by melting curves. TB Green^®^ Premix Ex Taq™ (no.RR420L, Takara) was used for mRNA qPCR. The reaction included one cycle of 95 °C for 2 min and forty cycles of 95 °C for 5 s, 60 °C for 30 s and 72 °C for 10 s. All experimental data were calculated by the 2^−ΔΔCt^ equation. ΔΔCt = (CT_target_ − CT_ref_) − Average (CT_target_ − CT_ref_), where target and ref represent the target gene and the reference gene, respectively. The normalized factor (NF) values of the reference gene and target gene were calculated by Excel 2020 software.

### 4.6. Western Blot (WB) Analysis 

Cells were also collected after transfection to extract proteins. The tail fat of six samples was collected to verify the protein expression of DDI1. A total of 0.1 g of sheep tail fat and cells were extracted. An amount of 1 mL of pre-cooling RIPA lysis buffer (25 mM Tris-Hcl pH 7.6, 150 mM NaCl, 1% NP-40, 1% sodium deoxycholate, 0.1% SDS, and 1 mM PMSF) was added to obtain the total proteins and measure the protein concentrations via the BCA method (Thermo Scientific, 23235, Waltham, MA, USA). Furthermore, 4× reduced sample buffers (Thermo Scientific, NP0007) were added and incubated at 95 °C for 5 min. The proteins (20 ug/lane) were separated by 4–12% SDS-PAGE, and 30–70 kDa of protein were removed to transfer onto a PVDF membrane (Millipore) with the Bio-Rad transferring system (Bio-Rad, 1703930). Furthermore, the membrane was sealed at room temperature for 1.5 h with 5% skim milk powder (Thermo Scientific, LP0031B). Anti-DDI (44 kDa, Proteintech, 13968-1-AP), Anti-PDGFD (57 kDa, Bioss, bs-5776R), Anti-PPAR-γ (58 kDa, Proteintech, 60127-1-Ig), anti-β-tubulin (50 kDa, Proteintech, 66240-1-Ig), and anti-β-actin (42 kDa, Bioss, bs-0061R) were diluted with 3% skim milk powder, then incubated on the PVDF membrane overnight at a temperature of 4 °C. The membrane was washed three times with TBST (Solarbio, T1085) for 8 min at a time. After that, the membrane was incubated with a secondary antibody (Abcam, ab6721) at room temperature for an hour. The membrane was washed three times with TBST for 8 min at a time. The reaction band was developed using enhanced chemiluminescence (Thermo Scientific, 32209), and the membrane was imaged with a Tanon 5200 imaging system (Tanon, Shanghai, China). The verification information for the commercial antibodies can be easily found on the manufacturer’s website. 

### 4.7. Statistical Analysis

Unpaired student’s *t*-test was used for two-group comparisons, and one-way ANOVA analysis was used for multiple-group comparisons by GraphPad Prism version 8.0.2 software (GraphPad Software, San Diego, CA, USA). All statistical tests were considered significant at * *p* < 0.05 and ** *p* < 0.01. And the data were also visualized in GraphPad Prism version 8.0.2 software (GraphPad Software, San Diego, CA, USA). The results are presented as a means ± SEM (standard error). The gray analysis was conducted in ImageJ software, using the equation IntDen_target protein_/IntDen_reference protein_ [50].

## 5. Conclusions

Our results showed that oar-miR-432 inhibits fat differentiation through the target gene *DDI1* in ovine preadipocytes. This study contributes to understanding the importance of oar-miR-432 inducing differentiation in preadipocytes. For the first time, this article details the mechanism of fat deposition via *DDI1* inhibition in ovine preadipocytes and *DDI1*’s co-expression with *PDGFD*. These results pave the way for further research on the adaptation mechanism by controlling sheep tail fat deposition as well as obesity-related diseases. 

## Figures and Tables

**Figure 1 ijms-24-11567-f001:**
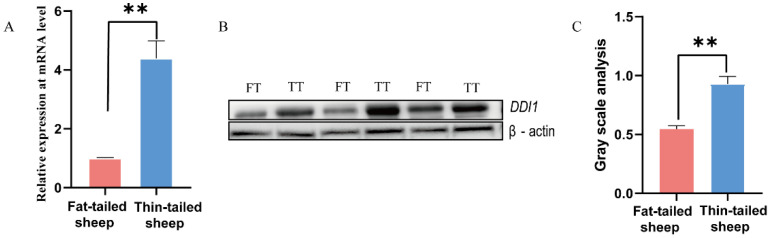
Expression of *DDI1* in tail fat tissues of fat-tailed sheep (FT) and thin-tailed sheep (TT). (**A**) RT–qPCR analysis of *DDI1* expression in the tail fat of fat-tailed and thin-tailed sheep. (**B**,**C**) Western blot analysis and gray value analysis of DDI1 in tail fat tissues of fat-tailed and thin-tailed sheep. The Western blot cut 30–70 kDa of protein to be transferred onto a PVDF membrane, ** *p* < 0.01.

**Figure 2 ijms-24-11567-f002:**
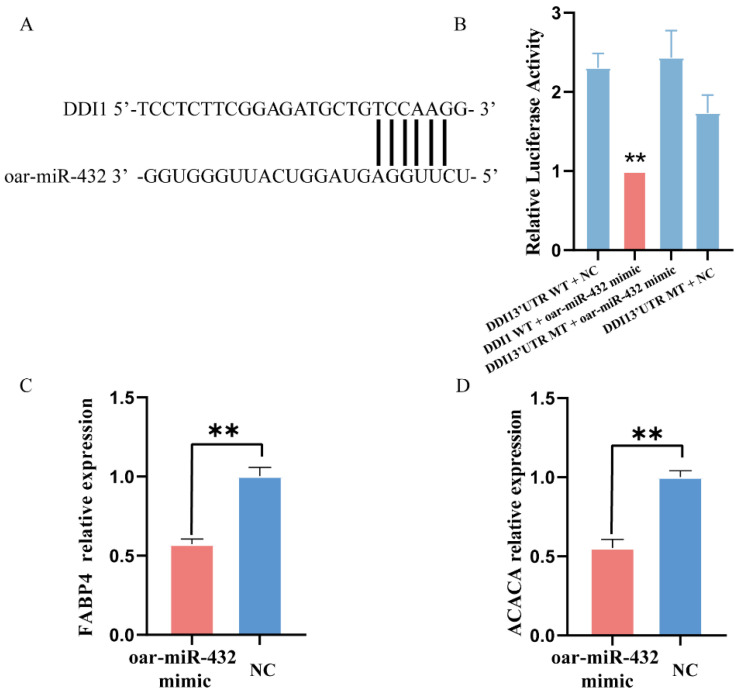
Oar-miR-432 directly target DDI1 3′ UTR. (**A**) A putative binding site of oar-miR-432 was found in *DDI1* 3′-UTR. (**B**) Dual luciferase reporter assays to detect the predicted target genes *DDI1* of oar-miR-432 (n = 4). Relative expression of *FABP4* (**C**) and *ACACA* (**D**) when the miR-432 mimic, the oar-miR-432 mimic and their NC were transfected into preadipocytes after four days. Data are presented as mean ± SEM (n = 3), ** *p* < 0.01.

**Figure 3 ijms-24-11567-f003:**
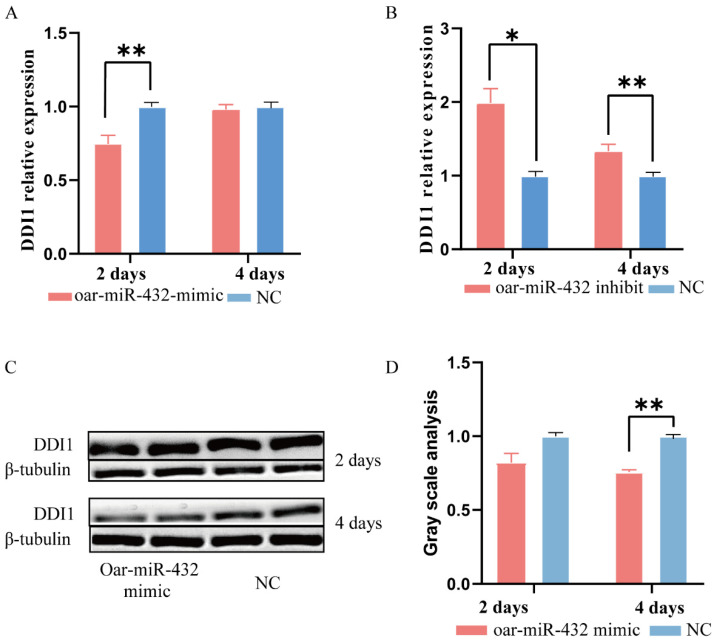
Oar-miR-432 negatively regulates the expression of *DDI1*. (**A**) The mRNA expression of *DDI1* induces differentiation after transfection of miR-432 mimics or NC into ovine preadipocytes after two and four days. (**B**) The mRNA expression of *DDI1* induces differentiation after transfection of the miR-432 inhibitor or NC into ovine preadipocytes after two and four days. (**C**,**D**) Western blot analysis and gray value analysis of DDI1 when the oar-miR-432 mimic and NC were transfected into preadipocytes after two and four days. The Western blot cut 30–70 kDa of protein to be transferred onto a PVDF membrane. Data are presented as means ± SEM (n = 3). ** *p* < 0.01 and * *p* < 0.05.

**Figure 4 ijms-24-11567-f004:**
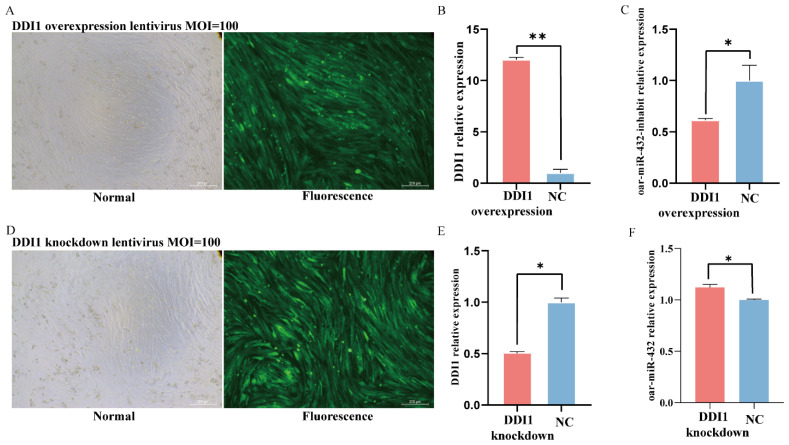
*DDI1* down-regular oar-miR-432 in ovine preadipocyte. (**A**) *DDI1* lentiviral overexpression transfected into ovine preadipocytes, MOI = 100. (**B**) The efficiency of *DDI1* overexpression. (**C**), the relative expression levels of oar-miR-432 after transfection with *DDI1* overexpression. (**D**) *DDI1* lentivirus knockdown transfected into ovine preadipocytes, MOI = 100. (**E**) The efficiency of *DDI1* knockdown. (**F**) The relative expression levels of oar-miR-432 after transfection with *DDI1* knockdown. Data are presented as mean ± SEM (n = 3). ** *p* < 0.01 and * *p* < 0.05.

**Figure 5 ijms-24-11567-f005:**
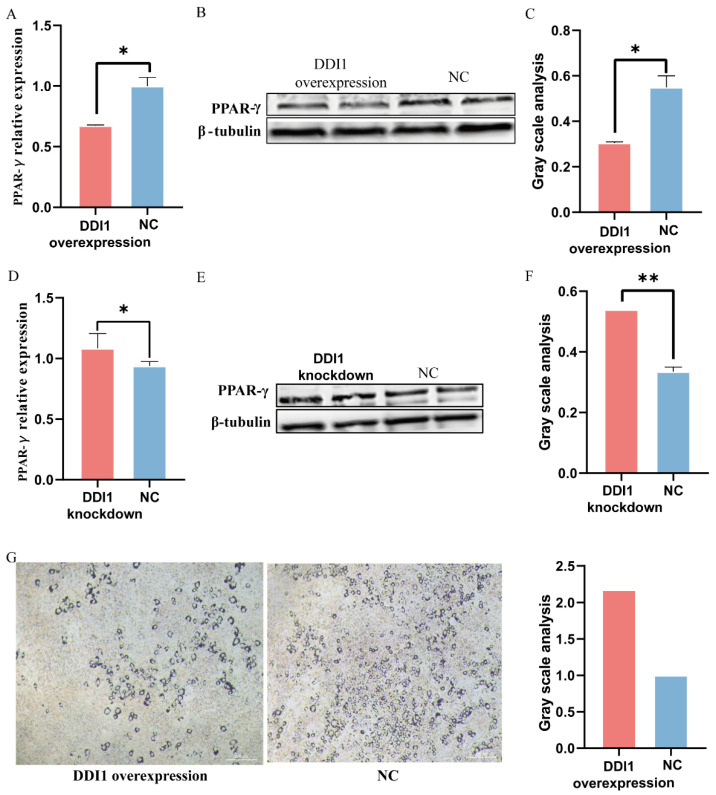
The function of *DDI1* regarding fat deposition in sheep tail preadipocyte. (**A**) The relative expression levels of *PPAR-γ* after transfection with lentivirus-induced *DDI1* overexpression. (**B**,**C**) Western blot analysis and gray value analysis of the protein expression of *PPAR-γ* after transfection with *DDI1* overexpression. (**D**) The relative expression levels of *PPAR-γ* after transfection with lentivirus-induced *DDI1* knockdown. (**E**,**F**) Western blot analysis and gray value analysis of the protein expression of *PPAR-γ* after transfection with lentivirus-induced *DDI1* knockdown. (**G**) Oil red O staining and gray value analysis in *DDI1* over-expressing sheep preadipocytes. Data are presented as mean ± SEM (n = 3). ** *p* < 0.01 and * *p* < 0.05.

**Figure 6 ijms-24-11567-f006:**
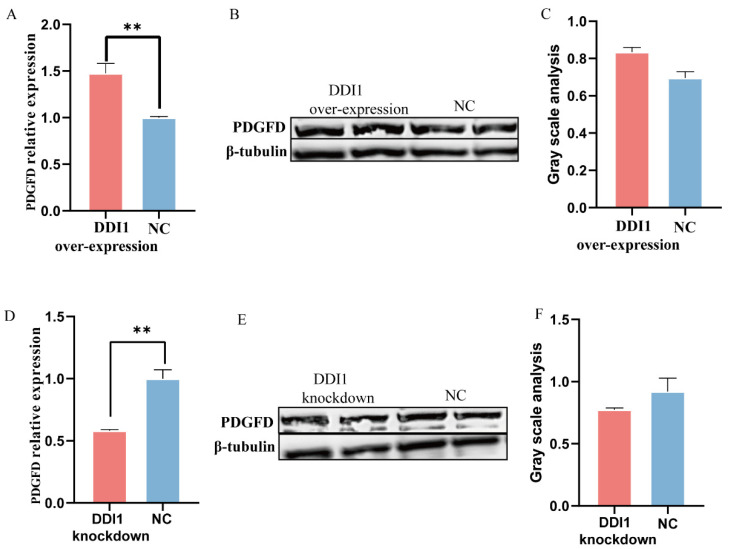
The expression of PDGFD influenced by DDI1 overexpression or knockdown. (**A**) The relative expression levels of *PDGFD* after transfection with overexpressing *DDI1*. (**B**,**C**) Western blot analysis and gray value analysis of *PDGFD* expression after transfection with overexpressing *DDI1*. (**D**) The relative expression levels of *PDGFD* after transfection with lentivirus-mediated knockdown *DDI1*. (**E**,**F**) Western blot analysis and gray value analysis of *PDGFD* expression after transfection with knockdown *DDI1*. Data are presented as mean ± SEM (n = 3), ** *p* < 0.01.

**Table 1 ijms-24-11567-t001:** Primers used in this study for RT-qPCR.

Gene	Prime (5′ to 3′)	Tm	GenBank
DDI1-F	CACAGTCTATTGCGTGCGGA	57.6	XM_012095299.4
DDI1-R	AGGAGCCCAGGGAGCAGTGA	57.2	
PDGFD-F	CTCAGGCGAGATGAGAGCAA	55.9	XM_004015965.5
PDGFD-R	TCCTTGTGTCCACACCATCG	57.7	
PPAR-γ-F	ATAAAGCGTCAGGGTTCCAC	56.3	NM_001100921.1
PPAR-γ-R	ATCCGACAGTTAAGATCACACC	54.0	
FABP4-F	AGCCACTTTCCTGGTAGCAA	56.5	NM_001114667.1
FABP4-R	ACACACGCCTGCTCTTTCTTA	56.5	
ACACA-F	GTTTCGTCTCCACCACCGAA	57.7	NM_001009256.1
ACACA-R	CTGACTCGTCAGTGAGACCG	57.8	
β-actin-F	CCAACCGTGAGAAGATGACC	56.5	NM_001009784.3
β-actin-R	CCCGAGGCGTACAGGGACAG	57.1	
oar-miR-432-RT	GTCGTATCCAGTGCAGGGTCCGAGGTATTCGCACTGGATACGACCCACCC	-	NR_107875.1
oar-miR-432-F	CGCGTCTTGGAGTAGGTCATT	58.5	
oar-5s-RT	GTCGTATCCAGTGCAGGGTCCGAGGTATTCGCACTGGATACGACAGCCTA	-	XR_003590556.1
oar-5s-F	TGGGAATACCGGGTGCTG	57.96	

## Data Availability

Not applicable.

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
