# Peer review of "The Effects of DDI1 on Inducing Differentiation in Ovine Preadipocytes via Oar-miR-432"

_ijms, 2023, doi:10.3390/ijms241411567_

Round 1

Reviewer 1 Report

The fat tail is a unique characteristic of sheep that represents energy reserves and is a complex adaptative mechanism of fat-tailed sheep to environmental stress. MicroRNA plays a significant role as regulators at the posttranscriptional level, and other studies have explained the molecular mechanisms of miRNA which regulate fat deposition in sheep tails, for example in Hu fat-tailed and Tibetan thin-tailed sheep. Among differentially expressed genes, oar-miR-432 was one of the most downregulated miRNAs, which targets about 712 genes, i.e., that at least this number of genes was predicted to be targeted by oar-miR-432, for example previous data reported that oar-miR-432 mimic transfected into preadipocytes resulted in increased expression of BMP2. In humans, BMP2 is a novel, depot-specific regulator of adipogenesis in human subcutaneous adipose tissue (AT). These adipocytes also strongly implicate BMP2-SMAD1/5/8 signalling in the determination examining multiple AT depots, of body fat distribution in humans and emphasise the importance of comparative studies in this field. Which consideration the authors made about the relationship with human studies?

DDI1, which is not DDl1, is not spelled out in the text. This is burdensome for the comprehension of the article. Is that DDI1 the proteasome receptor which has been previously identified both in Drosophila photoreceptor neurons and in human neuroblastoma cells in culture as a direct substrate of UBE3A, i.e., the ubiquitin E3 ligase?

Statistics must b thoroughly revised, as no rigorous test to infer statistics was reported. Therefore, in this circumstance, statistics cannot be accepted and differences between data non trustable.

Ethical issues. The Declaration reported in methods about ethical issues regarding farm animals and animal protection must be better detailed, as the reference to the indicated document is not appropriate and does not contain elements regarding ethical issues in treating with animals. So, the protocol cannot be associated with any regulative institution.

Authors have to check acronyms ed edit language typos.

Author Response

Comments and Suggestions for Authors

1.The fat tail is a unique characteristic of sheep that represents energy reserves and is a complex adaptative mechanism of fat-tailed sheep to environmental stress. MicroRNA plays a significant role as regulators at the posttranscriptional level, and other studies have explained the molecular mechanisms of miRNA which regulate fat deposition in sheep tails, for example in Hu fat-tailed and Tibetan thin-tailed sheep. Among differentially expressed genes, oar-miR-432 was one of the most downregulated miRNAs, which targets about 712 genes, i.e., that at least this number of genes was predicted to be targeted by oar-miR-432, for example previous data reported that oar-miR-432 mimic transfected into preadipocytes resulted in increased expression of BMP2. In humans, BMP2 is a novel, depot-specific regulator of adipogenesis in human subcutaneous adipose tissue (AT). These adipocytes also strongly implicate BMP2-SMAD1/5/8 signalling in the determination examining multiple AT depots, of body fat distribution in humans and emphasise the importance of comparative studies in this field. Which consideration the authors made about the relationship with human studies?

Reply: Thank you for your valuable comments. We have added the possible relationship between fat deposition in sheep tailed and human obesity in the discussion section. This study provides insights into the fat regulation mechanism in fat-tailed sheep and has important applications in animal breeding, as well as obesity-related human diseases. For example, in humans, when fat accumulates disproportionately in the viscera, rather than in the buttocks and hips, the risks of cardiovascular metabolic diseases such as coronary heart disease are increased (line 353-357).

BMP2 induces the directed differentiation of stem cells into white fat and influences fat deposition in sheep tail [45]. In humans, BMP2 is a novel depot-specific regulator of adipogenesis in subcutaneous adipose tissue [46]. BMP2 increases hyperplasia and hypertrophy of bovine subcutaneous preadipocytes via BMP/SMAD signaling [47]. These results suggest that BMP2 may have similar regulatory mechanisms for fat deposition in different species. (line 416-420)

  1. DDI1, which is not DDl1, is not spelled out in the text. This is burdensome for the comprehension of the article. Is that DDI1 the proteasome receptor which has been previously identified both in Drosophilaphotoreceptor neurons and in human neuroblastoma cells in culture as a direct substrate of UBE3A, i.e., the ubiquitin E3 ligase?

Reply: Thank you for your valuable and thoughtful comments. We have included DDI1-related information in the revised manuscript (line 55-58). DDI1 is a eukaryotic protein with a retroviral protease fold, which has been previously identified both in Drosophila photoreceptor neurons and in human neuroblastoma cells in culture as a direct substrate of the ubiquitin E3 ligase.

  1. Statistics must b thoroughly revised, as no rigorous test to infer statistics was reported. Therefore, in this circumstance, statistics cannot be accepted and differences between data non trustable.

Reply: Thank you for your suggestion. We reanalyzed all the data and plotted new figures.

In the original manuscript, all experimental data were calculated by the 2-ΔΔCt equation without a normalized process. Statistical analyses were performed in the ANOVA program of SPSS 20.0 software (IBM). (line 165-167)

In the revised manuscript, we reanalyzed these data by calculating normalized factor (NF) values.(line 510-511) Unpaired student's t-test was used for two-group comparisons and one-way ANOVA analysis was used for multiple-group comparisons by GraphPad Prism version 8.0.2 software (GraphPad Software, San Diego, CA, USA). (line 535-541)

  1. Ethical issues. The Declaration reported in methods about ethical issues regarding farm animals and animal protection must be better detailed, as the reference to the indicated document is not appropriate and does not contain elements regarding ethical issues in treating with animals. So, the protocol cannot be associated with any regulative institution.

Reply: Thank you for your suggestion. We have revised the ethical issues according to the opinions of the reviewers. We delete the protocol in the materials and methods part (line 94-97). In Declaration, we have modified the format of ethics statement according to the literature published in International Journal of Molecular Sciences journal.(line 556-559)

5.Comments on the Quality of English Language

Authors have to check acronyms ed edit language typos.

Reply: Thank you very much for discovering this error. We apologize for this grammatical problem and have corrected it based on your suggestions. In addition, we have asked native English editors to polish and modify the manuscript (https://www.mdpi.com/authors/english).

Reviewer 2 Report

In this manuscript, the authors have studied the effect of oar-miR-432 on fat differentiation.  

Results showed that oar-miR-432 inhibits DDI1 while the inhibition of oar-miR-432 increased DDI1. DDI1 negatively regulates fat deposition in ovine preadipocytes by inhibiting PPAR-gamma expression. Moreover, the expression of DDI1 influenced that of PDGFD.

The authors concluded that oar-miR-432 is important in inducing the differentiation of preadipocytes via DDI1 inhibition.

This is an interesting paper with the methods described in detail, the results attractively presented, and the conclusions are sound. However, before the manuscript is acceptable for publication the authors need to address the following points:

1.    It is not clearly indicated which results were obtained with preadipocytes and which with HEK293T cells. In addition, why were HEK293T cells used?

2.    DDI1, PDGFD, BMP2, etc should be abbreviated when first appeared, as NC (line 96), and MOI (line 117) as well.

3.    Legends of Figure 2 (line 230) and of Figure 6 (line 318); “*P<0.05” should be eliminated because single asterisks are not present in the figures.

4.    In Figure 3, the SEMs of NC are not shown.

5.    The sentence in lines 257-260 seems to refer to another Figure. In the legend of Figure 4, B and E were described with “The efficiency of DDI1 overexpression” and “The efficiency of DDI1 271 knockdown”.

6.    Line 263-264; It is not understood why this should be surprising given that DDI1 and oar-miR-432 appear to have an inverse correlation. Please revise this.

7.    In Figure S2, the oar-miR-432 mimic was indicated in the x-axis and not DDI1 overexpression as shown in the legend and in the text (p. 8 lane 285).

8.    Lanes 337-338; it is not clear how this sentence relates to the study.

9.    Lanes 340-341; Are DDI1 and fat transport related? 

10. Lines 349-351; it seems that a single sentence has been split into three sentences by adding full stops.

11. Lines 373-376; sentences do look not essential to the discussion.

English language should be revised

Author Response

Comments and Suggestions for Authors

In this manuscript, the authors have studied the effect of oar-miR-432 on fat differentiation.

Results showed that oar-miR-432 inhibits DDI1 while the inhibition of oar-miR-432 increased DDI1. DDI1 negatively regulates fat deposition in ovine preadipocytes by inhibiting PPAR-gamma expression. Moreover, the expression of DDI1 influenced that of PDGFD.

The authors concluded that oar-miR-432 is important in inducing the differentiation of preadipocytes via DDI1 inhibition.

This is an interesting paper with the methods described in detail, the results attractively presented, and the conclusions are sound. However, before the manuscript is acceptable for publication the authors need to address the following points:

  1. It is not clearly indicated which results were obtained with preadipocytes and which with HEK293T cells. In addition, why were HEK293T cells used?

Reply: Thank you for your careful work. Oar-miR-432 mimic, inhibit, DDI1 overexpression and knockdown were obtained with preadipocytes. (line 463)

The Dual-Luciferase Reporter Assay was obtained with HEK293T cells. We have added this information in the revised manuscript. (line 459-460)

HEK293T transfection efficiency is high, so the cardinality is large, thus eliminating the problem of individual differences caused by the transfection efficiency.

  1. DDI1, PDGFD, BMP2, etc should be abbreviated when first appeared, as NC (line 96), and MOI (line 117) as well.

Reply: Thank you for your careful work. We have added the full name of these abbreviations when they first appeared in the manuscript. (line 50-52, 209, 293, 369, 392-394, 447, 470)

  1. Legends of Figure 2 (line 230) and of Figure 6 (line 318); “*P<0.05” should be eliminated because single asterisks are not present in the figures.

Reply: Thank you for your careful work. We have deleted *P<0.05 in the legends of Figure 2 and Figure 6.(line 243, 338)

  1. In Figure 3, the SEMs of NC are not shown.

Reply: Thank you for your careful work. We have revised Figure 3 in the revised manuscript. (line 259)

  1. The sentence in lines 257-260 seems to refer to another Figure. In the legend of Figure 4, B and E were described with “The efficiency of DDI1 overexpression” and “The efficiency of DDI1 271 knockdown”.

Reply: We are sorry for our mistakes in the manuscript. We have corrected the error. DDI1 also enhanced the expression of oar-miR-432 after transfection of the lentivirus-induced overexpression DDI1 (Figure 4B) and suppressed the expression of oar-miR-432 after transfer of the lentivirus-induced DDI1 knockdown (Figure 4E). (Line272-275)

  1. Line 263-264; It is not understood why this should be surprising given that DDI1 and oar-miR-432 appear to have an inverse correlation. Please revise this.

Reply: Thank you for your valuable suggestion. We have rephrased this sentence in the revised manuscript. (line 278)

  1. In Figure S2, the oar-miR-432 mimic was indicated in the x-axis and not DDI1 overexpression as shown in the legend and in the text (p. 8 lane 285).

Reply: We are sorry for our mistakes in the manuscript. We have revised Figure S2 in the revised manuscript.

  1. Lanes 337-338; it is not clear how this sentence relates to the study.

Reply: Thank you for your valuable comment. We agree with your opinion. We have deleted this sentence in the revised manuscript. (line 358)

  1. Lanes 340-341; Are DDI1 and fat transport related?

Reply: Thank you for your comment. In our study, our results show that DDI1 inhibits ovine preadipocytes inducing differentiation. Whether DDI1 is related to fat transport needs further research. In the revised manuscript, we have deleted the confusing sentences. (line 359-361)

  1. Lines 349-351; it seems that a single sentence has been split into three sentences by adding full stops.

Reply: Thank you for your careful work. We have rewritten this sentence in the manuscript (line 369-373). WDTC1 serves as a substrate receptor for DDB1-CUL4-ROC1 (CRL4) E3 complexes, which are involved in transcriptional repression during adipogenesis.

  1. Lines 373-376; sentences do look not essential to the discussion.

Reply: Thanks for your valuable suggestions. We have moved these sentences to the other position and made appropriate modifications to make the article smoother. (line 396-409)

  1. Comments on the Quality of English Language

English language should be revised

Reply: Thank you very much for discovering this error. We apologize for this grammatical problem and have corrected it based on your suggestions. In addition, we have asked native English editors to polish and modify the manuscript (https://www.mdpi.com/authors/english).

Reviewer 3 Report

Jin et al reported an interesting finding regarding the role of oar-mir-432 and DD1 in adipocyte differentiation in the tail of sheeps.

I would like to address several points:

1. It would be advisable to use unified term for the sheep, whether using Hu sheep or fat-tailed sheep and Tibetan sheep or thin-tailed sheep to describe the results throughout the manuscript.

2. Authors reported that Oar-mir-432 mimic decreased FABP4 and ACACA, therefore, adipocyte differentiation was inhibited. Can you please provide other genes related to adipocyte differentiation, such as leptin or genes related to lipid droplet formation, or genes regulated by PPARg?

3. Authors stated that Oar-mir-432 mimic decreased the expression of DD1, Oar-mir-432 inhibition increased the expression of DD1 (Figure 3). Later, authors also wrote that DD1 regulated Oar-mir-432. Could you please explain more detail about this? Please elaborate more which one is the regulator.

4. For Figure 4. DDI1 down-regular oar-miR-432 in ovine preadipocyte, can you please provide the protein expression data from western blot?

5. Could you please calculate the differentiation rate that may be derived from figure 5G?

6. I would advice to move line 191-195 to the methods section.

I hope my points may be helpful to improve the manuscript.

Moderate editing is required. some grammatical errors were found in the manuscript.

Author Response

Comments and Suggestions for Authors

Jin et al reported an interesting finding regarding the role of oar-mir-432 and DD1 in adipocyte differentiation in the tail of sheep.

I would like to address several points:

  1. It would be advisable to use unified term for the sheep, whether using Hu sheep or fat-tailed sheep and Tibetan sheep or thin-tailed sheep to describe the results throughout the manuscript.

Reply: Thank you for your valuable comments. We have used a unified term for the sheep, fat-tailed sheep represent Hu sheep and thin-tailed sheep represent Tibetan sheep in the revised manuscript. (line 19, 80,197, 200, 206, 222-223, 343-344, 441)

  1. Authors reported that Oar-mir-432 mimic decreased FABP4 and ACACA, therefore, adipocyte differentiation was inhibited. Can you please provide other genes related to adipocyte differentiation, such as leptin or genes related to lipid droplet formation, or genes regulated by PPARg?

Reply: Thank you for your valuable comments. We have also used PPAR-γ to verify oar-miR-432 inhibits fat deposition in preadipocytes, and these results have been published (Jin, M., Fei, X., Li, T. et.al. Oar-miR-432 Regulates Fat Differentiation and Promotes the Expression of BMP2 in Ovine Preadipocytes. Front Genet, 2022, 13: 844747).

  1. Authors stated that Oar-mir-432 mimic decreased the expression of DD1, Oar-mir-432 inhibition increased the expression of DD1 (Figure 3). Later, authors also wrote that DD1 regulated Oar-mir-432. Could you please explain more detail about this? Please elaborate more which one is the regulator.

Reply: Oar-miR-432 directly interacts with the predicted target sites in 3’UTR of DDI1 and negatively regulates the expression of DDI1 in ovine preadipocytes. Because DDI1 inhibits ovine preadipocytes inducing differentiation, so maybe influence the expression of oar-miR-432. The regulator is oar-miR-432.

Previous studies have shown that miRNAs can target negative regulatory target genes, but nobody has looked at the effect of miRNAs after overexpression or knockdown of target genes. In the spirit of exploration, we overexpressed and knockdown DDI1 to verify the expression of miR-432. However, why the expression of miRNA is affected by the target gene still needs further research.

  1. For Figure 4. DDI1 down-regular oar-miR-432 in ovine preadipocyte, can you please provide the protein expression data from western blot?

Reply: Thank you for your suggestion. We agree with you that using western blot to further verify these results. Previously, genes of miRNAs were assumed to be incapable of encoding proteins. The biogenesis of miRNAs involves the processing of larger primary miRNAs (pri-miRNAs) into shorter pre-miRNAs, and the maturation of pre-miRNA to produce active miRNAs (Ha M, Kim VN. Regulation of microRNA biogenesis. Nat Rev Mol Cell Biol. 2014; 15:509–24.). However, recent studies have demonstrated that pri-miRNAs harbor short open reading frames that may encode regulatory peptides, termed miRNA-encoded peptides (Kang, M.; Tang, B.; Li, J.; et al. Identification of miPEP133 as a novel tumor-suppressor microprotein encoded by miR-34a pri-miRNA. Mol Cancer 2020, 19, 143.).

In our work, we did not find an ORF in miR-432 based on the existing literature and thus we didn’t implement WB.

  1. Could you please calculate the differentiation rate that may be derived from figure 5G?

Reply: Thank you for your valuable comments. We have calculated the differentiation rate from Figure 5g and added a picture in Figure 5g. Compared with NC, DDI1 overexpression increased the 2.17-fold-change differentiation rate in preadipocytes. (line 305-306,312)

  1. I would advise to move line 191-195 to the methods section.

Reply: Thank you for your valuable comments. We have moved line 441-443 to the methods section.

  1. Comments on the Quality of English Language

Moderate editing is required. some grammatical errors were found in the manuscript.

Reply: Thank you very much for discovering this error. We apologize for this grammatical problem and have corrected it based on your suggestions. In addition, we have asked native English editors to polish and modify the manuscript (https://www.mdpi.com/authors/english).

Round 2

Reviewer 1 Report

The authors fulfilled this Reviewer's recommendations

Reviewer 2 Report

This manuscript has been revised according to my suggestions. Now I am satisfied and I believe that It is acceptable for the pubblication.

Reviewer 3 Report

Authors have addresses all of my comments.